# Neglected Zone VII Extensor Tendons Reconstruction with a Palmaris Longus Tendon Autograft

**DOI:** 10.3390/medicina61020249

**Published:** 2025-02-01

**Authors:** Łukasz Wiktor, Ryszard Tomaszewski

**Affiliations:** 1Department of Trauma and Orthopaedic Surgery, Upper Silesian Children’s Health Centre, Medical University of Silesia, 40-752 Katowice, Poland; 2Department of Trauma and Orthopedic Surgery, ZSM Hospital, 41-500 Chorzów, Poland

**Keywords:** children, extensor tendon, tendon repair, palmaris longus

## Abstract

*Background:* This study reported a case of zone VII multiple neglected extensor tendons reconstruction with a palmaris longus tendon autograft in a 15-year-old boy 3 months after the initial trauma. *Case presentations:* Preoperative examinations revealed complete damage of the extensor carpi radialis longus (ECRL), extensor carpi radialis brevis (ECRB), abductor pollicis longus (APL), and partial injury of the extensor pollicis brevis (EPB). The extensor tendons were reconstructed with a palmaris longus tendon autograft combined with graft tunnel reconstruction within the scar at the level of the damaged retinaculum. After the surgical treatment, short immobilization and early rehabilitation were applied, providing passive sliding of the reconstructed tendon supplemented with actively mediated extension. *Results:* Despite the neglectful nature of the injury, surgical treatment and early postoperative rehabilitation resulted in an excellent functional outcome. At the follow-up visit, 6 months postoperative, the patient presented a full range of motion of the radiocarpal joint and thumb without any limitations on hand function. *Conclusions:* (1) Palmaris longus tendon autograft is a viable option for the treatment of multiple zone VII extensor tendon damage. (2) The combination of early passive motion and actively mediated extension provides tendon gliding and results in good functional outcomes for a hand with zone VII extensor tendon injury. (3) Ultrasound examination can evaluate early results and detect complications, mainly tendon/graft adhesions, after extensor tendon reconstruction surgery.

## 1. Introduction

Hand injuries constitute a significant major portion of all cases treated in hospital emergency departments and may reportedly constitute up to approximately 15% of all trauma cases [1].

Based on the single-center comparative analysis of pediatric tendon injuries in the hand compared with adults, these damages are much less common in children [2]. In the pediatric group, being male, the right hand, the extensor tendon, complete rupture, the middle finger, and glass injury predominated among tendon injuries [2]. Extensor injuries in zone VII were the least numerous and accounted for less than 2% [2].

The extensor tendons are located superficially beneath the skin and subcutaneous tissue, making them susceptible to injuries to the dorsal surface of the hand and forearm. Depending on the location of the injury, diagnosis is made through physical examination and various provocative tests supported by ultrasound or magnetic resonance scan. Furthermore, according to anatomical landmarks, injuries to the extensor tendons can be classified into eight damage zones, as shown in Table 1 [3]. The extensor tendon system is a complex interplay between an extrinsic and intrinsic extensor apparatus [4]. Extrinsic extensors arise from the dorsal forearm and traverse distally in superficial and deep compartments. The superficial compartment contains extensor carpi radialis longus (ECRL), extensor carpi radialis brevis (ECRB), extensor digitorum communis (EDC), extensor digiti minimi (EDM), and extensor carpi ulnaris (ECU). The deep compartment includes the abductor pollicis longus (APL), extensor pollicis brevis (EPB), extensor pollicis longus (EPL), and extensor indicis proprius (EIP). The intrinsic extensors consist of four dorsal interossei muscles, three palmar interossei muscles, and four lumbrical muscles, with tendons running palmar to the MCP axis and forming lateral bands on either side of each finger.

Due to the complicated anatomical structure and partial overlapping of the hand and finger extensor functions, damage to particular tendons at an early stage may be demanding to detect based on a physical examination, especially if the physician is not experienced in assessing the extensor apparatus. Extensor tendon injuries are a common and complex medical issue directing knowledge of anatomy, repair and reconstructive techniques, and rehabilitation protocols. Treatment can be nonoperative or operative, depending on the zone of injury with a high complication rate, and outcomes are generally less favorable for more distal injuries.

This paper aims to show the results of treating neglected multiple extensor tendon damage in zone VII using a palmaris longus (PL) tendon graft. Due to the low incidence of such injuries in children and the gap in the literature, this paper provides valuable practical information for physicians dealing with tendon injuries.

A 15.7-year-old boy was admitted to the Orthopedics Department 82 days after injury to the dorsal side of the left forearm. On the day of the injury, he was treated outside our hospital due to a deep cut wound in the distal part of the left forearm (discharge card data: surgical debridement and layered suturing of the wound with no tendon damage). Upon admission to the Department, a healed posttraumatic wound was found without any signs of inflammation. Noteworthy was the involuntary positioning of the left hand in palmar flexion and ulnar deviation with active wrist extension and hand deficiency, a deficit of radial deviation, and limited abduction of the left thumb. There were no signs of blood supply and innervation disorders distally to the injury. Preoperative examinations were supplemented with an MRI scan of the forearm, which revealed complete damage and retraction of the proximal tendon ends in the scope of extensor carpi radialis longus (ECRL), extensor carpi radialis brevis (ECRB), abductor pollicis longus (APL), and partial injury of the extensor pollicis brevis (EPB).

MRI scans showing the distal ends of the injured ECRL and ECRB tendons are shown in Figure 1. The extensor pollicis longus (EPL) tendon was preserved. After preparation, the patient underwent surgery. The procedure was performed under a tourniquet. The post-traumatic wound was widened proximally and distally to visualize and release scar tissue properly. After mobilization and resection of the tissue scar, the distal ends of the damaged radial extensors were visualized (Figure 2). The extent of the damage matched the MRI results. The proximal ends of the ECRL and ECRB formed a common scar and underwent retraction that did not allow mobilization (Figure 3).

After debriding ends, the tendon deficit was estimated at 4 cm for ECRB and 5 cm for ECRL. After removing the scar band around the damaged APL tendon and mobilizing the ends, the tendon defect was estimated at 4 cm. Partial EPB damage did not require repair. The extensor retinaculum was damaged for the second and third compartments covered by scar tissue. The first compartment was preserved, including EPB, but with a loss of APL. Due to the long-lasting forced positioning of the hand and partial damage to the extensor retinaculum, ulnar tucking of the undamaged extensor apparatus of the hand was found. The palmaris longus (PL) tendon, with a total length of 17 cm, was collected from separate microincisions on the palmar surface using a dedicated harvester. After surgical debridement, the graft was divided into three parts with lengths of 2 × 5 cm and 1 × 6 cm, respectively. The grafts prepared were then sewn into the distal ends of ECRL, ECRB, and APL using the Pulvertaft technique. The scar at the level of the damaged retinaculum was debrided, followed by surgically reconstructing separate graft tunnels. After placing the hand in an intermediate position, the PL grafts were sewn into the proximal extensor ends “overlapping” with a 1 cm margin (Figure 4). After the surgery, the limb was immobilized in a palmar splint in a dorsal flexion of approximately 20° for 2 weeks until the first follow-up. After confirming proper wound healing, the sutures were removed, and the splint was modified to ensure a neutral hand position. Two weeks after surgery, we started passive wrist and thumb motion to ensure the gliding of the reconstructed tendons and avoid potential adhesions. Four weeks after the procedure, the plaster splint was changed to an orthosis, holding the wrist in a neutral position, and actively mediated extension was added into the rehabilitation protocol. In Figure 5, we have shown longitudinal ultrasound sections of the reconstructed tendons, confirming the efficacy of the distal Pulvertaft sutures with no splitting of the tendon ends. Six weeks after surgery, the patient began intensive rehabilitation under the supervision of the outpatient rehabilitation department. Twelve weeks after the extensor reconstruction, during a follow-up, active wrist extension and radial wrist deviation were preserved with a deficit of 20° and 10°, respectively. The thumb abduction was maintained in the full range but with a mild limitation of thumb opposition. The patient was referred to additional rehabilitation combined with manual therapy and scar tissue mobilization. At the follow-up visit, 6 months after the reconstructive surgery, the patient presented a full range of motion: video documentation (Appendix A).

## 2. Discussion

Considering the anatomy and severity of the injury, it should be emphasized that wounds in the dorsal part of the wrist and the distal forearm are often associated with simultaneous damage to numerous tendons. Restoring the extensor continuity and suitable soft tissue coverage becomes crucial for obtaining proper extensor apparatus function. Regarding the tubular shape and the greater range of tendon sliding in zones VI to VIII, from a technical point of view, tendon repair becomes less demanding than in more distal zones. However, tendon injuries in zone VII require extensor retinaculum repair to prevent tendon bowstringing.

In zones VI to VIII, for isolated injuries, it is possible to perform a side-to-side tendon transfer using the adjacent uninjured tendon while maintaining the function of the hand. For example, ECRL can be transferred to the distal end of the damaged extensor to regain thumb or finger extension [5]. However, the abovementioned procedures become impossible for multiple tendon tears. If primary repair is impossible, especially in neglected cases like ours, the surgeon must be well versed in many divergent reconstructive techniques. The preference of the appropriate surgical method must be based on the location of the injury, tendon gap, and damage to the surrounding soft tissues. Secondary reconstructive techniques include intercalary grafting (auto or allograft), free tissue grafting (composite grafts), and tendon transfer for designated lesions [4]. According to Kochevar A. et al., impaired alignment of tendons in zones II to IV may ultimately lead to lower biomechanical extensors efficiency [6]. The accurate alignment of tendons in zone VII at the extensor retinaculum level seems equally important. In our case, we did not observe any weakening of extension strength or range of motion deficits, and interestingly, after restoring the correct hand position, the subluxation of the intact extensors observed intraoperatively was withdrawn. Due to the numerous tendon damage and neglect, we decided to perform a multiple bridge graft using the autologous palmaris longus tendon as the donor. The literature has widely described PL tendon bridge grafting as one of the most common intercalary grafts [7]. Preoperative validation of the presence of PL is crucial because it exhibits high variability, and its prevalence ranges between 1.5% and 63.9% [8]. In our case, the presence of PL was confirmed in a clinical examination, and its length was initially estimated using MRI scans. If there is no PL, toe extensors or plantaris tendon can be used as a donor. Damage to the extensors in zone VII is inextricably linked to simultaneous damage to the extensor retinaculum. Palmer et al. reported that the extensor retinaculum is indispensable for the extension mechanism of the hand [9].

In our case, the repair of the damaged retinaculum consisted of creating two separate canals in the surrounding scar tissue at the spot of the second and third anatomical compartments to ensure good graft adhesion without conflict during sliding motion. This approach stays in line with Kawakatsu M. et al. reports, which emphasized that it is important to reconstruct only the extensor retinaculum, not the entire sliding surface [10]. Unlike the case described by the above authors, our patient did not require further tenolysis, probably because we introduced gentle passive movements from the third week after surgery. We avoided the necessity to release adhesions, which can also be explained by the fact that intraoperatively, during radical passive flexion and extension of the hand at the level of the reconstructed retinaculum, only the PL graft was sliding beneath without conflict with the level of suturing. A considerable issue during tendon reconstructive procedures is determining the appropriate graft tension. In the case of isolated extensor reconstruction, techniques for estimating graft tension are known, such as those described by Kamoi F. et al., who treated 20 patients with EPL damage with good results [11]. They assumed that the tension of the reconstructed EPL should be adjusted so that the center of the distal edge of the thumbnail is elevated 2 cm above the operation table. The situation is much more complex in the case of multiple tendon reconstruction. In our patient, the tension of the ECRB and ECRL grafts was determined to achieve a neutral hand position in both the flexion/extension and the ulnar/radial deviation plane. The tension of the APL graft was an intraoperative dilemma, which was sewn in with minimal tension, not guided by the position of the thumb. This may translate into afterward problems with rehabilitation in thumb opposition. Postoperative rehabilitation for extensor tendon damage in zones V to VII has historically relied on the immobilization of the hand for 4 to 6 weeks after surgery. From the literature, in zones V to VII, complications can arise with tendon adhesions, joint contracture, prolonged rehabilitation, and limitations of hand function. Extensor tendon rehabilitation providing passive or dynamically assisted sliding of the reconstructed tendon, the same as an actively mediated extension, significantly improves the quality of results compared to static splinting [8,12,13,14]. In our case, the introduction of the passive wrist and hand movements from the third week after surgery assisted active extension from the fifth week after surgery, and the patient’s involvement in the rehabilitation process resulted in a very good functional outcome. Ultrasound examination (UE) should focus on assessing early results and detecting complications, mainly tendon adhesions. UE also allows for managing rehabilitation progress and assessing the sliding movement of the repaired tendon or tendon graft. In our case, UE was used to confirm the sliding movement of the PL grafts during passive extension movements at the early rehabilitation stage and to exclude splitting of the tendon ends.

Extensor tendon injury treatment requires understanding the complex anatomy and a working knowledge of the reconstructive techniques.

An orthopedic surgeon must thoroughly evaluate each wound to the dorsal surface of the forearm and hand for damage to the extensor tendons to avoid overlooking tendon harm. In uncertain cases, surgical revision of the traumatic wound should be considered to assess the state of the extensor tendons.

It is worth emphasizing that primary repair procedures are significantly effortless than secondary reconstructive techniques.

## 3. Conclusions

Palmaris longus tendon autograft is a viable option for the treatment of multiple zone VII extensor tendon damage.The combination of early passive motion and actively mediated extension provides tendon gliding and results in good functional outcomes for a hand with zone VII extensor tendon injury.Ultrasound examination can evaluate early results and detect complications, mainly tendon/graft adhesions, after extensor tendon reconstruction surgery.

## Figures and Tables

**Figure 1 medicina-61-00249-f001:**
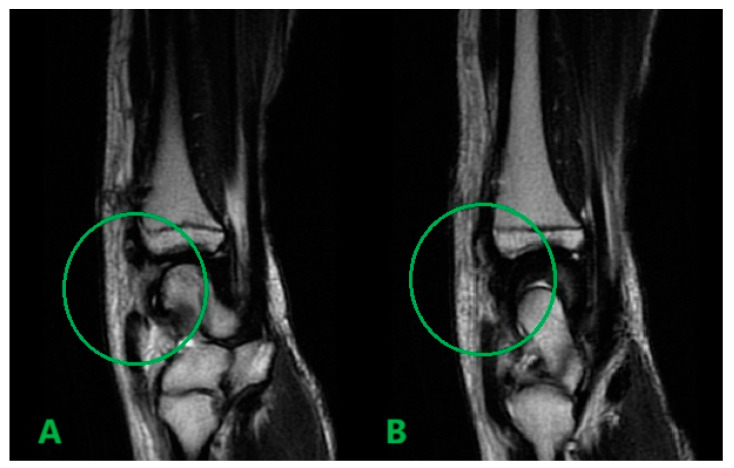
Preoperative sagittal T2 MRI scans showing the distal ends of the injured ECRL (**A**) and ECRB (**B**) tendons.

**Figure 2 medicina-61-00249-f002:**
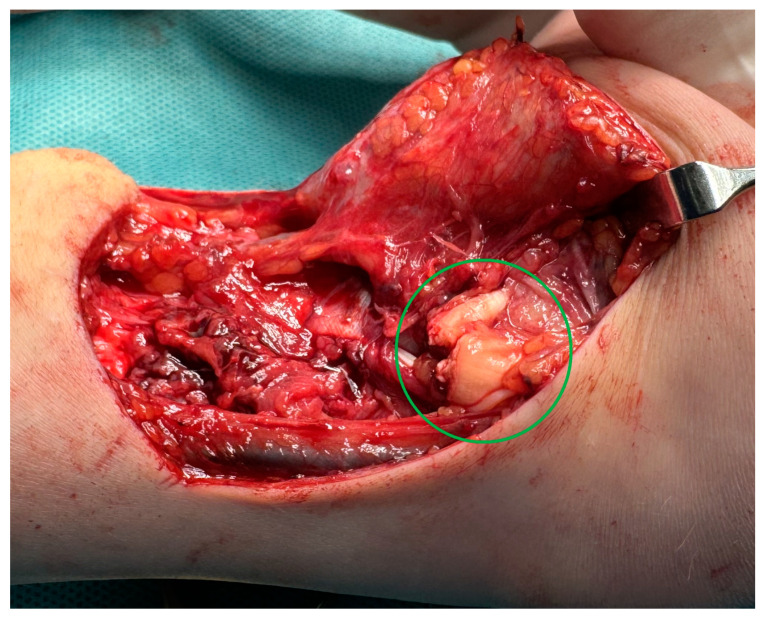
An intraoperative image showing the distal ends of the damaged ECRL and ECRB tendon.

**Figure 3 medicina-61-00249-f003:**
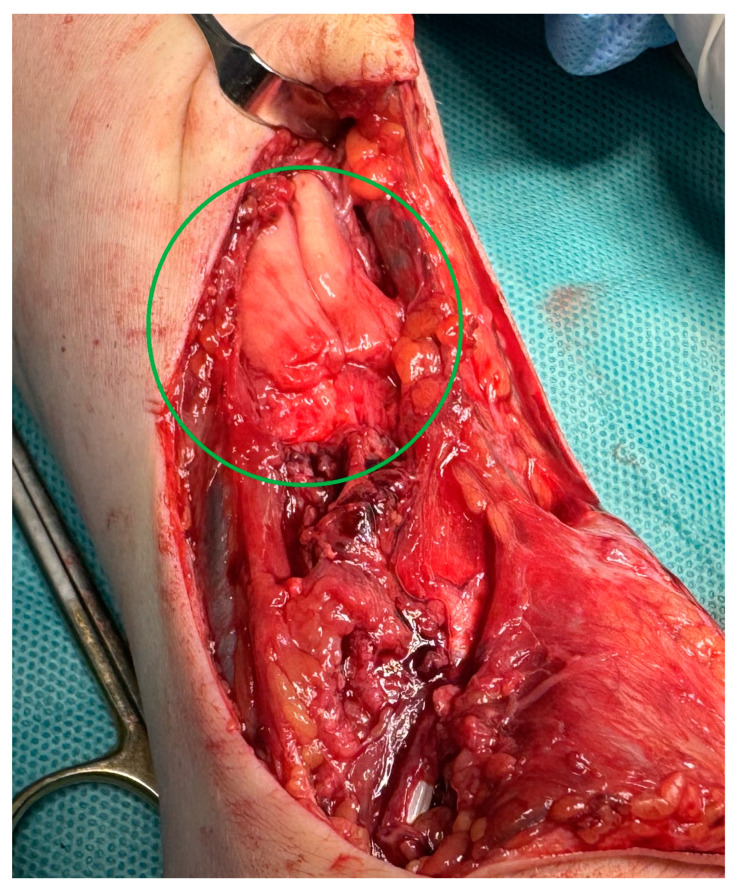
An intraoperative image showing the proximal ends of the damaged ECRL and ECRB tendon formed a common scar.

**Figure 4 medicina-61-00249-f004:**
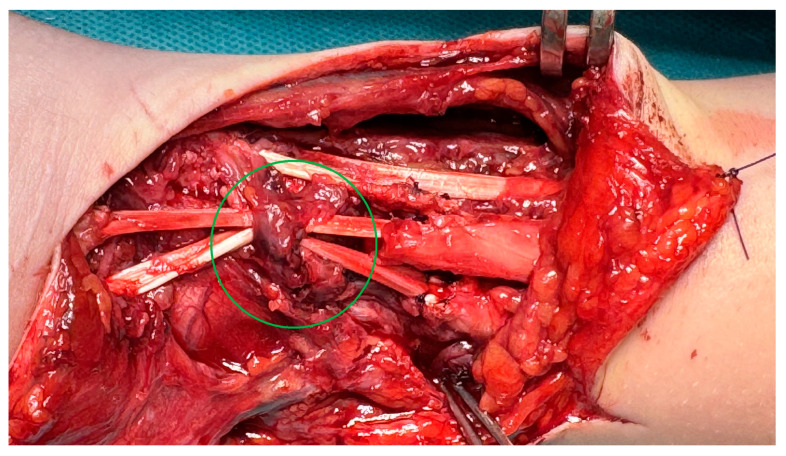
An intraoperative image after PL graft reconstruction of damaged ECRL, ECRB, and APL tendon. The rebuilt canal of the second and third extensor compartments is marked.

**Figure 5 medicina-61-00249-f005:**
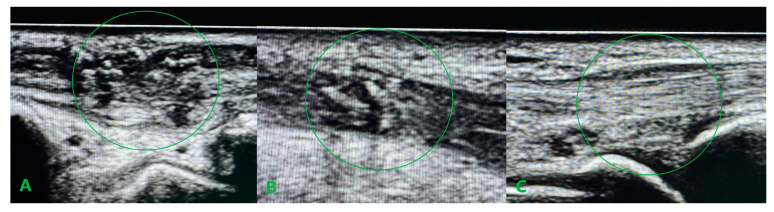
Longitudinal ultrasound sections of the reconstructed tendons six weeks after surgery confirming the efficacy of the distal Pulvertaft sutures. APL (**A**), ECRL (**B**), and ECRB (**C**). The most suitable ultrasound graft integration for ECRB tendon.

**Table 1 medicina-61-00249-t001:** Zones of extensor tendon injuries.

Zone	Level of Damage
I	distal to or at the DIP joint of the fingers and IP joint of the thumb (mallet finger)
II	over middle phalanx or proximal phalanx of thumb
III	over the PIP joint of digit or MCP joint of thumb (boutonniere deformity)
IV	over the proximal phalanx of digit or metacarpal of thumb
V	over the MCP joint of digit or CMC joint of thumb
VI	over the metacarpal
VII	at the wrist joint
VIII	at the distal forearm

## Data Availability

The original contributions presented in the study are included in the article/Appendix A; further inquiries can be directed to the corresponding authors.

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
