# Peer review of "Neglected Zone VII Extensor Tendons Reconstruction with a Palmaris Longus Tendon Autograft"

_medicina, 2025, doi:10.3390/medicina61020249_

Round 1
Reviewer 1 Report
Comments and Suggestions for Authors
Thanks to the authors for the interesting case.
Abstract: too brief. Please add the correct sections (background, case presentations, discussion, conclusion) to provide a clear state of the study. Abstract should offer an overview of the study: it is not acceptable in this way.
Introduction: sometimes unclear. Please improve this section by providing a clearer explanation of why such lesions cannot always be detected early.
Case Presentation: Well-structured.
Discussion: Overall well-written. What recommendations do the authors have for surgeons to accurately and promptly identify these neglected lesions?
Author Response
Author's Reply to the Review Report (Reviewer 1)
Dear reviewer.
We would like to kindly thank you for your time spent reviewing our manuscript “Neglected zone VII extensor tendons reconstruction with a palmaris longus tendon autograft. Case report’’. We appreciate all your valuable comments of our work. We would like to emphasize that all revisions made were marked up so changes can be easily viewed. Below We have included responses to the comments.
Best regards.
Łukasz Wiktor
Abstract: too brief. Please add the correct sections (background, case presentations, discussion, conclusion) to provide a clear state of the study. Abstract should offer an overview of the study: it is not acceptable in this way.
We have improved the Abstract, as recommended.
We hope that the changes will be satisfactory.
Introduction: sometimes unclear. Please improve this section by providing a clearer explanation of why such lesions cannot always be detected early.
We have improved the introduction section, as recommended. We hope that reorganizing this section will provide clarity. We have added the appropriate comment as recommended.
Case Presentation: Well-structured.
We appreciate all your comments.
Discussion: Overall well-written. What recommendations do the authors have for surgeons to accurately and promptly identify these neglected lesions?
We have added the appropriate comment as recommended.
Reviewer 2 Report
Comments and Suggestions for Authors
I would like to thank the authors for this valuable report. This case report contains valuable data on the reconstruction of missed hand injuries.
It is erroneous to present the general anatomical data in such exhaustive detail in the introduction and subsequently reiterate it in the table format. The table, in and of itself, is inadequate for this purpose, as it does not serve to provide any additional information. Please revise
At the end of the introduction, it would be advisable to incorporate a paragraph that underscores the significance of the case and its potential contributions.
The sharing of the case is adequate, and it is further supported by the incorporation of visual aids. The management and dissemination of the information are commendable.
The discussion was adequate, with salient points emphasised, and the conclusion was well-supported. The references are up to date and used appropriately.
The spelling of the study is adequate.
The figures and text complement each other well.
Following the completion of the minor revision, the manuscript will be suitable for publication.
Author Response
Author's Reply to the Review Report (Reviewer 2)
Dear reviewer.
We would like to kindly thank you for your time spent reviewing our manuscript “Neglected zone VII extensor tendons reconstruction with a palmaris longus tendon autograft. Case report’’. We appreciate all your valuable comments of our work. We would like to emphasize that all revisions made were marked up so changes can be easily viewed. Below We have included responses to the comments.
Best regards.
Łukasz Wiktor
It is erroneous to present the general anatomical data in such exhaustive detail in the introduction and subsequently reiterate it in the table format. The table, in and of itself, is inadequate for this purpose, as it does not serve to provide any additional information. Please revise
At the end of the introduction, it would be advisable to incorporate a paragraph that underscores the significance of the case and its potential contributions.
We have improved the introduction section, as recommended. We hope that reorganizing this section will provide clarity. We have added the appropriate comment as recommended. We have shortened the description of the anatomy of the extensor tendons. However, we believe the information in this section will facilitate the reader's interpretation and analysis of the case, especially for practitioners who do not deal with extensor injuries daily.
The sharing of the case is adequate, and it is further supported by the incorporation of visual aids. The management and dissemination of the information are commendable.
We appreciate all your comments.
The discussion was adequate, with salient points emphasised, and the conclusion was well-supported. The references are up to date and used appropriately.
We appreciate all your comments.
The spelling of the study is adequate.
We appreciate all your comments.
The figures and text complement each other well.
We appreciate all your comments.
Round 2
Reviewer 1 Report
Comments and Suggestions for Authors
I have carefully read the manuscript. Now it is more complete and clearer. Thanks to the authors for the corrections.